# Awareness and perceptions of Long COVID among people in the REACT programme: Early insights from a pilot interview study

Emily Cooper[1][☺][*], Adam Lound[1][☺], Christina J. Atchison[1], Matthew Whitaker[2,3], Caroline Eccles[4], Graham S. Cooke[5,6,7], Paul Elliott[2,3,6,7], Helen Ward[1,6,7,8]

1 Patient Experience Research Centre, School of Public Health, Imperial College London, London, United Kingdom, 2 School of Public Health, Imperial College London, London, United Kingdom, 3 MRC Centre for Environment and Health, Imperial College London, London, United Kingdom, 4 REACT-Long Covid Public Advisory Group, Patient Experience Research Centre, School of Public Health, Imperial College London, London, United Kingdom, 5 Department of Infectious Disease, Imperial College London, London, United Kingdom, 6 Imperial College Healthcare NHS Trust, London, United Kingdom, 7 National Institute for Health Research Imperial Biomedical Research Centre, London, United Kingdom, 8 MRC Centre for Global Infectious Disease Analysis and Abdul Latif Jameel Institute for Disease and Emergency Analytics, Imperial College London, London, United Kingdom

☺ These authors contributed equally to this work.
* emily.cooper@imperial.ac.uk

**Data Availability Statement:** There are ethical or legal restrictions on sharing a de-identified data set. The full study data cannot be made publicly available due to: • The sensitive nature of this data

## Abstract

### Background

Long COVID is a patient-made term describing new or persistent symptoms experienced following SARS-CoV-2 infection. The Real-time Assessment of Community Transmission-Long COVID (REACT-LC) study aims to understand variation in experiences following infection, and to identify biological, social, and environmental factors associated with Long COVID. We undertook a pilot interview study to inform the design, recruitment approach, and topic guide for the REACT-LC qualitative study. We sought to gain initial insights into the experience and attribution of new or persistent symptoms and the awareness or perceived applicability of the term Long COVID.

### Methods

People were invited to REACT-LC assessment centres if they had taken part in REACT, a random community-based prevalence study, and had a documented history of SARS-CoV-2 infection. We invited people from REACT-LC assessment centres who had reported experiencing persistent symptoms for more than 12 weeks to take part in an interview. We conducted face to face and online semi-structured interviews which were transcribed and analysed using Thematic Analysis.

### Results

We interviewed 13 participants (6 female, 7 male, median age 31). Participants reported a wide variation in both new and persistent symptoms which were often fluctuating or unpredictable in nature. Some participants were confident about the link between their persistent

(this is a small pilot study meaning there could be easy identification of participants), • The agreements and procedures governing usage and sharing of the collected dataset established by the study sponsors (UKRI/Medical Research Council and the National Institute for Health Research), research institution (Imperial College London) and Research Ethics Board (South-Central Berkshire B Research Ethics Committee (IRAS ID: 298404, REC Ref 21/SC/0134)). • The privacy protection and the nature of the consent obtained from participants which ensured confidentiality and anonymity. However, de-identified participant data used in the present study may be made available to investigators who complete the following steps: 1) present a study proposal that has received approval from an independent research committee or research ethics board; 2) provide a data request for review by the study principal investigators (PIs) and co-investigators; 3) following approval of the request, execute a data-sharing agreement between the investigators and the Study PIs. Study proposals and data access requests for elements of raw data may be sent to the REACT Data Access Committee (contact via react.access@imperial.ac. uk).

**Funding:** This work was supported by the NIHR/ UKRI [grant number COV-LT-0040]. HW is a National Institute for Health Research (NIHR) Senior Investigator and acknowledges support from NIHR Biomedical Research Centre of Imperial College NHS Trust, NIHR School of Public Health Research, NIHR Applied Research Collaborative Northwest London and the Abdul Latif Jameel Institute for Disease and Emergency Analytics. GSC is supported by a National Institute for Health Research (NIHR) Professorship. PE is Director of the MRC Centre for Environment and Health (MR/ L01341X/1, MR/S019669/1). PE acknowledges support from the NIHR Imperial Biomedical Research Centre and the NIHR HPRUs in Chemical and Radiation Threats and Hazards and in Environmental Exposures and Health, the British Heart Foundation Centre for Research Excellence at Imperial College London (RE/18/4/34215), Health Data Research UK (HDR UK) and the UK Dementia Research Institute at Imperial (MC_PC_17114). The principal investigator (PE) received the following awards: This study forms part of REACT Long COVID. We also acknowledge support from the NIHR Imperial Biomedical Research Centre, The Huo Family Foundation and the Department of Health and Social Care. The funders had no role in study design, data collection and analysis, decision to publish, or preparation of the manuscript.

symptoms and COVID-19; however, others were unclear about the underlying cause of symptoms or felt that the impact of public health measures (such as lockdowns) played a role. We found differences in awareness and perceived applicability of the term Long COVID.

## Conclusion

This pilot has informed the design, recruitment approach and topic guide for our qualitative study. It offers preliminary insights into the varied experiences of people living with persistent symptoms including differences in symptom attribution and perceived applicability of the term Long COVID. This variation shows the value of recruiting from a nationally representative sample of participants who are experiencing persistent symptoms.

## Introduction

The term Long COVID is now widely used to describe the ongoing ill-health experienced by a significant minority of people who have had SARS-CoV-2 infection [1]. Some organisations use different terms including post-COVID condition and post-acute covid syndrome, but Long COVID is the preferred term of patient groups. The term was coined by those with lived experience of persistent symptoms in the first months of the pandemic [2]. In the first half of 2020 little attention was paid to the emergence of longer-term symptoms, with surveillance focused on numbers of cases, hospital admissions and deaths. Much of the political and clinical focus was on reducing transmission and the impact of acute COVID-19 through public health measures including lockdowns. However, the messaging at the time, of a dichotomous disease profile of 'mild or severe', did not match the experience of many people who came together in online groups to share their stories [3]. Group members reported a range of symptoms and lack of recovery months later, even in those who experienced minimal symptoms at the initial stage of infection. The challenges faced by this growing community were compounded by the lack of COVID-19 testing to confirm and validate symptoms, and the focus of most research on people who were hospitalised with COVID-19.

The lack of recognition was the catalyst for, and focus of, many Long COVID support groups led by patients. These included people with 'professionally validated forms of authority', such as clinicians, academics and public representatives suffering from Long COVID themselves, who sought to raise awareness and prompt the call for 'recognition, research and rehabilitation' [3–5]. One early group was Body Politic on SLACK, which had an international reach and provided distinct spaces for specific demographic groups or people with particular types of symptoms [6]. The Patient-Led Research Collaborative was formed by members of the Body Politic group and produced the first survey of prolonged symptoms following COVID-19 infection in May 2020 [7]. In the UK, groups such as the Long COVID Support group on Facebook or the Long COVID SOS website emerged to provide more local support [8, 9].

Many Long COVID studies have recruited participants from these online support groups, social media or through specific clinical services for people with Long COVID [10]. While the support groups have been open to all, there will be many people and groups who are not included. Research into other chronic condition support groups has shown that taking the step to join can be difficult, with some feeling a diagnosis is needed to legitimise membership [11, 12]. People who are experiencing new or persistent symptoms following COVID-19 infection

**Competing interests:** The authors have declared that no competing interests exist.

may have limited awareness of Long COVID and this may reduce the likelihood that they seek out support groups or clinical assessment. Furthermore, the demographics of these support groups and social media communities appear to be skewed in terms of gender, ethnicity and age. For example, in their study of persistent COVID-19 symptoms, Davis et al distributed a survey via patient support groups and social media and found that the majority of respondents were white (85.3%), women (78.9%), aged between 30–59 (80.1%). They acknowledge that their findings 'may not be representative of the entire Long COVID population or their experiences' [1]. Further research drawing on more representative samples is essential to get a broader picture of who is being affected by Long COVID and how this impacts their lives.

The REACT (Real-time Assessment of Community Transmission) study involves repeated random cross-sectional samples of the population in England to quantify prevalence of virus in the community by RT-PCR (REACT-1), and to estimate the prevalence of IgG anti-SARS-CoV-2 antibody positivity based on a self-administered lateral flow immunoassay (LFIA) test (REACT-2) [13]. Taking this nationally representative sample as a framework for recruitment, the REACT-Long COVID (REACT-LC) study aims to identify the genetic, biological, social and environmental pathways that underpin progression to Long COVID following COVID-19. One component is a qualitative interview study to explore variations in experiences of people with new and persistent symptoms after infection with SARS-CoV-2 who were not hospitalised. We undertook a pilot study to inform the design of the larger qualitative study and to identify preliminary insights into the experience of new and persistent symptoms after COVID-19 and explore awareness and perceptions of the term 'Long COVID'.

## Methods

### Sample and recruitment

The pilot took place in the REACT-LC assessment centres between May and July 2021. People were invited to one of these assessment clinics across England if they had taken part in REACT and had a documented history of SARS-CoV-2 infection, were not hospitalised, and who may or may not have reported persistent symptoms. A total of 10,525 people attended these REACT-LC assessment centres where they underwent clinical tests and had samples taken for multiple biological investigation. The aim of these studies is to characterise genetic and downstream biological pathways responsible for individual differences in response to SARS-CoV-2 infection.

Participants were eligible to take part in the pilot study if they were attending the REACT-LC clinical assessment centre in West London where the research team is based and reported persistent or new symptoms (> 12 weeks) following COVID-19. This was a convenience sampling approach. We undertook the pilot study at a time when the assessment centres were prioritising recruitment of people aged 18–34 years old, allowing us to reach a younger cohort than those often included in Long COVID studies.

EC attended an assessment centre in West London, and with the help of the clinic research nurses and reception staff offered those who had persistent symptoms the opportunity to take part in a research interview immediately after their clinical assessment. Initial uptake was slow due to limited interviewer availability, a minority of people reporting persistent symptoms, participants being unable to stay for an interview and the challenges of social distancing within the clinical setting. The exact number of people who declined to participate are unknown as only those who 'opted in' were introduced to the researchers. The recruitment approach was subsequently altered to allow centre staff to consent eligible participants to be contacted by EC and AL for remote interviews (over phone or video) at a time convenient to them. We

undertook a total of 13 interviews with nine participants aged 18 to 34 years and four aged 35 to 44 years old.

## Topic guide

The topic guide was designed in collaboration with the REACT-LC Public Advisory Group (PAG). An initial outline was developed by EC, guided by the research questions and informed by current literature on Long COVID. This outline was then shared with and edited by PAG members individually, as well as during a series of group video calls. The pilot guide aimed to follow the patient journey from pre-COVID-19 health, through initial (acute) COVID-19 symptom experiences to then discuss any symptoms that persisted, or new symptoms that appeared, after 12 weeks. The guide also covered access to and availability of Long COVID support and views on Long COVID as a term. The advisory group suggested key areas of focus such as symptom fluctuation and factors that promoted or prevented recovery. The semi-structured style of the guide provided broad topics for discussion using prompts and probes to be used at the interviewer's discretion. The topic guide can be found in S1 Appendix.

## Interviews

Semi-structured interviews were carried out by EC and AL either face to face in the assessment centre or remotely by telephone or using video conferencing software (Zoom or Microsoft (MS) Teams) depending on participant preference. Interviews lasted between 19 and 70 minutes (mean 40 minutes). With participant consent, interviews were recorded using a digital recorder or using the built-in recording function on Zoom/MS Teams. After the interview was completed the audio recording was saved to a secure folder at Imperial College London and deleted from all devices. The audio recordings were sent for transcription to an external provider (UK Transcription). The transcripts were then downloaded back to the secure folder. The researchers kept field notes following interviews to support analysis.

## Analysis

An initial thematic analysis of the pilot interviews was carried out in NVivo. Transcripts were inductively coded by EC and AL and a consensus on the final codes was made through a process of comparison of the researcher's codebooks [14]. The content of the codes were then reviewed by the researchers, and descriptive topic summaries were developed through a process of detection [15]. Preliminary findings were not discussed with participants but were instead reviewed by the public advisory group. For this pilot study we drew on the experiences on a limited number of participants and undertook an initial thematic analysis to produce our current findings. These findings are descriptive and therefor are not presented as major or minor themes. While this study provides insights into experiences of living with persistent symptoms following Covid we do not claim to have reached data saturation and will endeavour to explore a more diverse range of voices in our main study.

## Patient and public involvement

The REACT-LC PAG is made up of nine members with experience of living with, or caring for someone, with Long COVID. Initial contributions included the development of the recruitment material and topic guide. A detailed summary of our pilot findings and planned approach to sampling for the main qualitative study were presented to the PAG for feedback. One PAG member (CE) undertook a review of this paper and is listed as a co-author.

### Reflexivity statement

EC is a white, female research assistant with a background in qualitative research and anthropology. AL is white, male research physiotherapist with a background in clinical practice specialising in respiratory medicine. EC and AL both have experience and training in qualitative methods. No relationship was established between researchers and participants prior to study commencement. Participants were given a brief explanation of EC and AL's role as qualitative researchers within the REACT-LC study and the overall goals of the project.

The authors were aware of how their own position may affect the study design, analysis and interpretation of the findings. Both researchers were aware of the current literature and social media discussions regarding Long Covid but neither had lived experience of the condition. We anticipated that participants would share a wide range of experiences in terms of symptom type, severity and impact but did not have pre-conceived ideas regarding potential findings or interpretations. The researchers worked closely with the REACT-LC PAG to support the analysis and reporting of our findings.

### Governance

This study received ethical approval from the South-Central Berkshire B Research Ethics Committee (IRAS ID: 298404, REC Ref 21/SC/0134) and all participants provided informed consent to be interviewed.

This manuscript is written in accordance with the Standards for Reporting Qualitative Research which can be viewed in S1 Checklist.

## Results

Thirteen people were interviewed between May and July 2021. The sample included 6 women and 7 men, aged between 18 and 44 years (median 31 years). Participants were from a range of ethnic groups with 8 being of white ethnicity. All had experienced persistent symptoms of COVID-19 for more than 12 weeks. Three had received a positive RT-PCR test and ten had a positive antibody test.

Below we report on our preliminary findings related to how participants experienced persistent symptoms and their perception of the cause of these symptoms. We also present findings related to their understanding, awareness, and views on the term Long COVID.

### Symptom experience

Participants reported a wide range of ongoing symptoms following the acute phase of COVID-19, with some reporting symptoms persisting well beyond the initial illness Others reported the development of new symptoms in the weeks and months following infection.

The symptoms that persisted tended to appear at some point during the initial illness and continued even after the individual felt they had largely recovered from the worst of the acute phase. These persistent symptoms included breathlessness, fatigue, headaches, loss of taste and smell, a lack of energy and for one participant, a continued temperature.

> *I had a temperature for, like, I would say, probably, three weeks. It wasn't as intense as the first four days. Then, I would still get days where my temperature would go really high, and that was quite debilitating.* (AL1, Female, 35–39)

The trajectory of symptoms varied and often fluctuated, were sometimes unpredictable ('volatile' or 'random' and others were 'cyclical'). One participant told a story of initial

recovery and tentatively restarting work, social, and physical activity only to experience a significant relapse in symptoms:

> *I was feeling more or less fine there. And then I started to push. So, I started to run a bit, because I was feeling better. . .But you know, like having pizza, more like a normal life. Like it's the weekend, it's sunny, you have a beer, you order pizza. And I started to run. And that was the worst mistake ever. That gave me a super big relapse. And then, in October, I went basically downhill again. So, my symptoms came back with revenge.* (EC7, Female, 30–34)

New symptoms, experienced once the acute illness had passed, were diverse in presentation with little consistency between individuals. Some of these new symptoms seemed similar to those experienced with the initial illness, such as a sore throat, temperature, weakness or headache, but they had appeared at a later point for some participants. For others, new symptoms were more distinct and included a variety of things such as loss of coordination, weight change, pins and needles, vision changes and chest pain among others. Sometimes the onset of such symptoms could be months after the initial infection:

> *Then on one particular day, it could have been a month, it might have been two after, but a long time after all the severe things, I remember just having the thing that everyone describes as like the rocks pressing down on your chest. Just one day of having that the most severe. . .if it happened again I probably would have taken myself to the hospital but it did only happen once really bad.* (EC3, Male, 30–34)

For some, these new symptoms included flaring up of pre-existing conditions, or were similar to symptoms they had experienced before COVID-19, making attribution uncertain. Menstrual changes were reported by a number of participants. One reported having periods which were 'really, really painful' and another described her symptoms getting worse during her period:

> *So, by end of April, I was feeling more or less good. Then, my period came and it was like all my symptoms started to be really, really bad. I woke up with my heart beating really, really fast. So, I called my GP and he referred me to go to the hospital.* (EC7, Female, 30–34)

Breathlessness and fatigue featured in participant descriptions of both persistent and new symptoms. The severity of breathlessness varied, with one participant needing to use his asthma inhaler more frequently after running, and another participant was so breathless they could no longer use the shower. Fatigue was a major concern with one person describing, 'waking up more tired than I went to bed' and another highlighting the compromises that are needed to manage fatigue:

> *So I guess the big things were fatigue. . . my partner has actually banned the 'T' word–we call it–in my house, me saying, "I'm tired," because he's like, "You say it all the time." (Laughter) So yes, I'm really tired. Normal activities that I would have done before that would be simple are exhausting. I have to put all my energy into work rather than anything else, anything social. Working out, exercise and things like that have really fallen by the wayside.* (EC14, Female, 35–39)

Participants were asked to describe a 'good day' and a 'bad day' to explore the extent of the fluctuation and the varying impact on their day to day lives:

*A good day I can smell my coffee, my toast, and I wake up before my alarm. I can have a full day on my feet, and at the end of the day, I've got energy to meet with a friend. So, that's a good day, like what it used to be, full of energy. A bad day, everything just takes a little bit extra energy. So, mentally as well, and emotionally, everything is just quite difficult to engage, or I feel a little bit distracted. I struggle at work being on my feet the whole day* (AL1, Female, 35–39)

## Perception of symptom cause

During the interviews participants were asked to describe their thoughts and feelings about their new or persistent symptoms. Some questioned whether these symptoms were related to COVID at all, instead suggesting other social or environmental determinants including lockdown and related changes could be responsible.

*Well, no. Yes, yes, well, I have constant headaches, that is true, but I don't know if that is because of COVID or just because I'm stressed, it is psychological. But it is very weird and sometimes it appears in the right-hand side of my head and then goes up and then to the left. Some days I lose entire days because I can't focus, but I don't know if that is COVID.* (EC4, Male, 30–34)

One participant reported how he was struggling to exercise since having COVID earlier in the year but that this was perhaps related to a lack of routine:

*I think that, because on my days off I would have enough energy to go to the gym, and just feel more motivated, but I don't have that at the moment. I'm just doing what I need to do, and then just rest. Yes, I don't know if it's because I need to get back into a routine. I don't know if lockdown messed that routine up, and I just need to get back into a routine, when I just need to get into things moving again. I don't know.* (AL1, Female, 35–39)

The effects of the pandemic and lockdowns over the past year were cited as a potential cause of symptoms. One participant highlighted how their mental health had been affected, leaving them feeling emotional due to a lack of control. Another suggested that she initially suspected her coordination and dizziness symptoms were due to being stuck indoors during lockdown (however, a lack of improvement since being outside again has led her to think it is something else).

Those participants who described more certainty that their new or persistent symptoms were COVID related had views on possible mechanisms driving their symptoms. Participants' understanding of the cause was determined by how the symptoms (or cluster of symptoms) manifested, what information they were able to glean from their own research or medical investigations, and what treatments they felt worked for them. For example, a participant experienced one-sided muscle weakness along with brain fog, blurred vision and loss of smell, which made her think of a neurological cause. However, she later explained that she was trying treatments related to a broad range of body systems and found some particularly helpful, despite not having been prescribed them:

*And I have found that the Vitamin B12 is really making something. I don't know if maybe my blood tests show that I was not deficient, but I have a feeling that maybe that was not totally right. Because when I have it, I feel basically normal, I even forget about COVID.* (EC7, Female, 30–34)

A participant who was suffering from non-specific chest pains expressed confusion about the cause of his pain. He could not tell whether it was his heart or his lungs due to the side of the body the pain was radiating from, and he was uncertain if there were in fact two separate kinds of pain, with different causes. On one occasion it appeared as a sharp intense pain then other times as gradual and less severe. He reported having multiple investigations including a chest X-ray in which clinicians reported 'they didn't see anything' leaving him with no further insight into the cause of his pain.

## Understanding and awareness and relevance of the term Long COVID

In the interviews we asked participants about COVID-19 including whether they were experiencing any persistent symptoms. We tried to avoid using the term 'Long COVID' in the interview at this point, although they were likely to be aware that the main study was called REACT-Long COVID. However, later in the interview we asked specifically about their knowledge and awareness of Long COVID although some had already referenced it when speaking about their symptom experience.

Those who didn't know much about Long COVID suggested it hadn't 'crossed their mind' as a possible diagnosis despite their persistent symptoms. Some did not view Long COVID as a distinct illness, for example, seeing these persistent symptoms resulting from the body still dealing with the consequences of the acute phase. Others described the ongoing symptoms as a natural part of the recovery process, which they saw as taking more time compared to other conditions, such as the flu.

*Patients who are still not recovering, they're still dealing with a lot of the symptoms. Their body is still taking a toll from what they've been through, and it's just affecting their day-to-day activities.* (EC6, Female, 24–29)

Some reported that they only found out about Long COVID through a clinician's suggestion, which led them to read more about the condition, but with the amount people reported carrying out their own research varying between participants. Receiving a potential diagnosis gave some participants a sense of vindication, confirming what they were experiencing was real and giving them a sense of control. For one participant however, the diagnosis was 'devastating' because there was no information about recovery or a cure.

A number of participants mentioned being made aware of Long COVID in the media and seeing the similarities to their own symptoms. However, for one person this meant seeing only the worst cases of Long COVID, which were much more severe than anything they had experienced. Witnessing severe cases of Long COVID, such as those depicted in the media, or experienced by family members, led some participants to feel that their own persistent symptoms were 'not severe enough' to count. One participant reflected on whether the term 'Long COVID' applied to them:

*So, I feel like it probably doesn't apply to me, because I feel like my symptoms, or whatever I have left, is not as severe as what other people are going through.* (AL1, Female, 35–39)

Confusion about what should be classed as Long COVID was widespread. Some felt that they were not sure if their symptoms 'fit' with the normal Long COVID symptoms because they were too 'niche' or that they were different to the more widely discussed symptoms like fatigue or breathlessness. One participant felt that the current definition was too 'wishy washy' to understand:

*Yes, so it's a bit wishy-washy, to me anyway, but also, I think everyone is feeling that they've got slightly different as well. It's not all A, B and C and that's what everybody is feeling. My kind of understanding is, "Oh, some people are getting headaches, some people still can't taste anything and some people. . ." You know? So it's one of those things. It's a bit of a broad term, isn't it? (EC14, Female, 35–39)*

Some shared a view that their experience did not fit with what would be commonly defined as Long COVID, however they also reported that they did not know the medical definition. It is important to acknowledge the interview effect on whether a person thinks they have Long COVID. One participant stated they had not thought of their symptoms as being Long COVID until they had been asked the questions for the interview:

*After talking to you, actually, yes, I was like, 'Hold on, really I could have had it at that [whole] time, I didn't really think about it in a way'. (EC1, Male, 30–34)*

## Views on the term Long COVID

Opinions on the term 'Long COVID' were mixed. Some liked the term and felt that it was 'sensible' and 'perfectly appropriate'; for example, using the term talking to a person in the street they felt people would understand what it meant. Others had no strong feelings, stating they had no objection or weren't really concerned by the specific name or label.

*Probably I would say tell it to anyone in the street and they would be like, "Yes, I get what Long COVID is. I would rather that term than say, chronic COVID, that would be a lot more worrying, we would be like, "Oh God, do I have a disease in me," or something like that.* (EC14, Female, 35–39)

One participant stated that the connotations of COVID as being chronic (and therefore being 'forever') would create a stress that could hinder her potential recovery. For her it was important to believe that she will, eventually, recover.

Those who disliked the term Long COVID felt it sounded as if they still had COVID. This was seen to be confusing for the public and that it could create fear, particularly in those who were shielding (it could be inferred that this was related to a fear of ongoing transmissibility).

One participant felt that the initial debates about Long COVID in the media had tarnished the term to some extent, leading some people to believe it was 'not a real thing'. Creating a new name for the condition, rather than using a term developed out of necessity, was put forward as a potential solution. Some of those who disliked the term Long COVID suggested the alternative label of 'post-COVID' which they felt emphasised that it is experienced *after* the initial infection:

*Yes, the Long COVID name to me, feels like you've still got COVID; it's like a long version of it, like you're still suffering from COVID. That's how it comes across to me. I always say post COVID, personally, just because usually when talking about things like post-op, you would say, not long-op. (EC12, Female, 24–29)*

## Discussion

### Summary and implications of our findings

In this pilot study we sought to gain preliminary insights into the experiences of those with new or persistent symptoms following COVID-19 infection. Participants experienced a diverse

range of symptoms that either persisted, or developed some time after, their initial SARS-CoV-2 infection. It was reported that multiple body systems were affected, and severity varied widely, with some participants reporting that persistent symptoms were significantly more debilitating than the early infection itself. A key challenge reported by participants was the unpredictable or fluctuating nature of their symptoms. Our findings on symptom experience echo the voices of those living with persistent symptoms [6, 8, 9] and concurrent research [16, 17].

Our pilot findings offer new insights into perspectives of those living with persistent symptoms which have been underexplored. We uncovered differing explanations of symptom cause and attribution. Some participants were confident there was a causal link between their current symptoms and COVID-19 based on an awareness of others with shared experiences, their own interpretation of existing scientific research, or confirmation by a medical professional. Others expressed doubt about whether COVID-19 was the actual cause of their symptoms. Alternative explanations offered by participants related to the physical and emotional impact of the pandemic.

We also found that not all participants were familiar with the term 'Long COVID' and some did not feel it applied to them or their symptoms. Some participants suggested their symptoms were related to a prolonged yet typical recovery from COVID-19 infection, rather than a distinct illness. Others were aware of Long COVID but felt that their symptoms did not fit the diagnosis or were not severe enough to count.

This variation in symptom attribution and understanding of the term Long COVID has implications for recruitment approaches and warrants deeper exploration in future research. We did not directly recruit from support groups or social media which have been used as a key source of participants for Long COVID symptom and qualitative studies [18]. Those we interviewed were part of the REACT study, experienced persistent symptoms and had chosen to participate in the clinical strand of the Long COVID study. However, they were not recruited based on being diagnosed with or believing themselves to have Long COVID. Our naïve assumption was that participants would associate what they were experiencing with the term Long COVID but through conducting the interviews it quickly became apparent that participants interpreted their symptoms in several different ways.

By including only those who identify as having Long COVID in studies we might fail to gain an insight into the perspectives and experiences of those who have new, unexplained or persistent symptoms but who do not recognise their condition as being included within the Long COVID category. These findings may also have impact beyond research. Those who do not recognise or identify with the term 'Long COVID' may not access healthcare services for investigation and management of their persistent symptoms or may not seek peer support through Long COVID groups.

In the absence of certainty around the 'biomedical facts' of this emerging illness there is potential for conflict around how Long COVID is understood, framed and categorised. Callard and Perego suggest the term is 'unstable' and highlight that it is crucial that 'Long COVID patients with different disease experiences and pathways are included in deliberations over terminologies used for long-term symptoms/illness' [3]. In our main qualitative study, we will further explore the insights raised in the preliminary findings of this pilot.

## Planned changes to our main qualitative study

The primary aim of this study was to inform the design and conduct of our main qualitative study. Undertaking this pilot has helped us identify key areas for improvement, particularly in terms of sampling and recruitment strategies, the topic guide and analytic approach. A full

overview of how this pilot has informed our main qualitative study design can be found in S2 Appendix.

## Sampling and recruitment strategies

The study sample was drawn from wider REACT cohort, which was itself a representative sample of the population of England. However, the limitations of our sampling approach and the small-scale nature of this pilot meant we were not able to reflect the full diversity of age and ethnicity in the current study. Furthermore, recruitment took place in a single geographic location and participants needed to be well enough to physically attend a REACT-LC assessment clinic, so we are likely to have not captured those experiencing more debilitating Long COVID symptoms.

In the main study we aim to recruit a larger (up to 60 participants) and more representative sample. We will broaden our recruitment approach to include those who have attended REACT-LC assessment centres across the country and additionally draw on respondents to a REACT-LC questionnaire, which will be sent to more than 200,000 REACT participants. We will purposively sample for diversity of experience and demographic background. This will be particularly important for ensuring the ethnic diversity of the sample. The lack of ethnic diversity in health research participation, and the existing barriers has been widely documented [19]. A recent rapid review highlighted a lack of diversity in Long COVID qualitative studies and encouraged researchers to prioritise recruitment of 'patients from diverse social and ethnic backgrounds to better represent the demographics of wider society' [17].

## Topic guide and analytic approach

Undertaking pilot interviews provided insights into how the topic guide in the main study could be improved. We identified a number of gaps and found that the structure of the guide did not fit with the natural trajectory of the narratives shared by participants. One significant improvement will be the mapping of change relative to participant baseline in 2019. We will use their pre-pandemic experience as an anchor to prompt and probe participants on what has changed and how their symptoms (and other contextual pandemic factors) have impacted these elements of their lifestyle. The topic guide will be adjusted to allow for more in-depth exploration into the lived experience of new and persistent symptoms including treatment and recovery as well as new topics such as changes to menstrual cycles, sex and intimacy. We will conduct more in-depth and explanatory analysis of the data using reflexive Thematic Analysis in the main qualitative study to provide insights beyond basic descriptions of different experiences [20]. We anticipate that our approach will be informed by social science literature to allow us to draw on theory related to how illness is constructed, contested and stigmatized when examining the evolving meaning of the term 'Long COVID' [21–23].

## Conclusion

We undertook a pilot to inform the design of our main REACT-LC qualitative study. This pilot provided preliminary insights into the experiences of new and persistent symptoms following COVID-19 infection. Even within this small sample we identified significant variation in symptom type and severity as well as differences in awareness or perceived applicability of the term Long COVID. The findings also show a variation in participant's attribution of ongoing symptoms.

These insights highlight the value of including a range of people in Long COVID studies, not just those already accessing services and support groups. By recruiting beyond these groups, research might reach those who do not seek, or are not offered, support and treatment

for their persistent symptoms. This pilot study, along with feedback from our advisors with lived experience of Long COVID, has shaped the development of the main qualitative study. We have broadened our recruitment approach and will undertake purposive sampling based on variations in experience and demographics. Additional improvements include restructuring the topic guide and taking a more in-depth and reflexive approach to analysis. We hope other social science researchers may find our insights helpful in designing their own Long COVID studies.

## Supporting information

**S1 Checklist. COREQ checklist.**
(PDF)

**S1 Appendix. Topic guide.**
(DOCX)

**S2 Appendix. Limitations of the pilot and how they will be addressed in the main study.**
(DOCX)

## Acknowledgments

Thank you to all the participants who agreed to share their experiences with us. We gratefully acknowledge the support provided by our PAG members, Graham Blakoe, Patricia Veres, Grace Bazunu, Hema Swami, Kimberly Bennet, and Jordan Jenkins.

## Author Contributions

**Conceptualization:** Emily Cooper, Adam Lound, Christina J. Atchison, Paul Elliott, Helen Ward.

**Data curation:** Emily Cooper, Adam Lound.

**Formal analysis:** Emily Cooper, Adam Lound.

**Funding acquisition:** Graham S. Cooke, Paul Elliott, Helen Ward.

**Investigation:** Emily Cooper, Adam Lound.

**Methodology:** Emily Cooper, Adam Lound, Paul Elliott, Helen Ward.

**Project administration:** Emily Cooper, Adam Lound, Paul Elliott, Helen Ward.

**Resources:** Paul Elliott, Helen Ward.

**Supervision:** Christina J. Atchison, Helen Ward.

**Validation:** Christina J. Atchison, Helen Ward.

**Writing – original draft:** Emily Cooper, Adam Lound.

**Writing – review & editing:** Emily Cooper, Adam Lound, Christina J. Atchison, Matthew Whitaker, Caroline Eccles, Graham S. Cooke, Paul Elliott, Helen Ward.

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
