## [Decision Letter · Decision Letter 0]

28 Nov 2022

PONE-D-22-18885Awareness and perceptions of Long COVID among people in the REACT programme: early insights from a pilot interview studyPLOS ONE

Dear Dr. Cooper,

Thank you for submitting your manuscript to PLOS ONE. After careful consideration, we feel that it has merit but does not fully meet PLOS ONE’s publication criteria as it currently stands. Therefore, we invite you to submit a revised version of the manuscript that addresses the points raised during the review process.

ACADEMIC EDITOR:

Please address the following comments:

The following minor revisions are suggested to improve the manuscript:

1. The introduction provides a valuable account of patient activism during the pandemic, and provides an informative background and history of “Long COVID”’s identification. The points citing tweets from Elisa Perego (references 2 and 3, line 7-13) may be better supported by the publications she co-authored on the topic. See Callard, F., & Perego, E. (2021). How and why patients made Long Covid. Social science & medicine, 268, 113426. (https://www.sciencedirect.com/science/article/pii/S0277953620306456) and Munblit, D., O'Hara, M. E., Akrami, A., Perego, E., Olliaro, P., & Needham, D. M. (2022). Long COVID: aiming for a consensus. The Lancet Respiratory Medicine. (https://www.sciencedirect.com/science/article/pii/S2213260022001357)

2. The authors’ reflections on diversity and inclusion strengthen this work. Given the impetus in the introduction to recruit a diverse sample and the reported lack of diversity in the discussion, it would be interesting to know the diversity of the REACT cohort the authors were able to select from. This could be expanded on in the discussion – was there a diverse sample in the REACT study but the pilot interview study failed to recruit a representative selection, or was the present study limited by an already non-diverse REACT cohort?

3. Consider following reporting standards by including the COREQ (COnsolidated criteria for REporting Qualitative research) Checklist in your supplementary information. This is a checklist of items that should be included in reports of qualitative research (https://cdn.elsevier.com/promis_misc/ISSM_COREQ_Checklist.pdf).

4. In the Methods/Analysis section the topic summaries are described as being “identified”. In Braun & Clarke’s 2019 paper on TA (https://www.tandfonline.com/doi/full/10.1080/2159676X.2019.1628806) they explain: “Themes are analytic outputs developed through and from the creative labour of our coding. They reflect considerable analytic ‘work,’ and are actively created by the researcher at the intersection of data, analytic process and subjectivity. Themes do not passively emerge from either data or coding; they are not ‘in’ the data, waiting to be identified and retrieved by the researcher.” Consider amending to terms like ‘developed’ ‘constructed’ or ‘generated’ to capture this process.

Overall this is an excellent manuscript and I recommend this for publication with minor revisions. I 

Please insert comments here and delete this placeholder text when finished. Be sure to:Indicate which changes you require for acceptance versus which changes you recommendAddress any conflicts between the reviews so that it's clear which advice the authors should followProvide specific feedback from your evaluation of the manuscriptPlease ensure that your decision is justified on PLOS ONE’s publication criteria and not, for example, on novelty or perceived impact.

We look forward to receiving your revised manuscript.

Kind regards,

Mohsen Abbasi-Kangevari

Academic Editor

PLOS ONE

https://journals.plos.org/plosone/s/fileid=ba62/PLOSOne_formatting_sample_title_authors_affiliations.pdf.

Reviewers' comments:

Reviewer's Responses to Questions

**Comments to the Author**

1. Is the manuscript technically sound, and do the data support the conclusions?

Reviewer #1: Yes

Reviewer #2: Yes

2. Has the statistical analysis been performed appropriately and rigorously? 

Reviewer #1: N/A

Reviewer #2: Yes

3. Have the authors made all data underlying the findings in their manuscript fully available?

Reviewer #1: Yes

Reviewer #2: No

4. Is the manuscript presented in an intelligible fashion and written in standard English?

Reviewer #1: Yes

Reviewer #2: Yes

5. Review Comments to the Author

Reviewer #1: This is a pilot qualitative study to inform a more larger qualitative study. the authors look have gained information about how to design further study. this has also been reflected in the findings and in the conclusion. I recommend the manuscript be accepted.

Reviewer #2: Peer Review of PONE-D-22-18885

“Awareness and perceptions of Long COVID among people in the REACT programme: early insights from a pilot interview study”

Thank you for the opportunity to review this manuscript, which presents the findings from pilot interviews with participants from a wider COVID-19 study (REACT), to explore their experiences and gather insights ahead of a larger qualitative study. Given the magnitude of COVID-19, patients with “Long COVID” have been epistemically overlooked. This work adds value to our knowledge of patients’ experiences and perceptions of the term “Long COVID” with important implications for future research.

The manuscript has several key strengths:

- Meaningful patient involvement work; the high level of patient involvement is particularly appropriate given the patient-driven movement towards identifying and coining the term “Long COVID” on which this study is based.

- It is very well written, with clear aims and rationale for conducting the work. The results are well presented, with sophisticated integration of quotes into the text and thoughtful interpretation.

- Transparency in providing detail on how the recruitment drive for people aged 18-34 enabled the authors to reach a young cohort, how the recruitment strategy was altered, and for providing the topic guide and lessons from pilot study in their supplementary materials.

The following minor revisions are suggested to improve the manuscript:

1. The introduction provides a valuable account of patient activism during the pandemic, and provides an informative background and history of “Long COVID”’s identification. The points citing tweets from Elisa Perego (references 2 and 3, line 7-13) may be better supported by the publications she co-authored on the topic. See Callard, F., & Perego, E. (2021). How and why patients made Long Covid. Social science & medicine, 268, 113426. (https://www.sciencedirect.com/science/article/pii/S0277953620306456) and Munblit, D., O'Hara, M. E., Akrami, A., Perego, E., Olliaro, P., & Needham, D. M. (2022). Long COVID: aiming for a consensus. The Lancet Respiratory Medicine. (https://www.sciencedirect.com/science/article/pii/S2213260022001357)

2. The authors’ reflections on diversity and inclusion strengthen this work. Given the impetus in the introduction to recruit a diverse sample and the reported lack of diversity in the discussion, it would be interesting to know the diversity of the REACT cohort the authors were able to select from. This could be expanded on in the discussion – was there a diverse sample in the REACT study but the pilot interview study failed to recruit a representative selection, or was the present study limited by an already non-diverse REACT cohort?

3. Consider following reporting standards by including the COREQ (COnsolidated criteria for REporting Qualitative research) Checklist in your supplementary information. This is a checklist of items that should be included in reports of qualitative research (https://cdn.elsevier.com/promis_misc/ISSM_COREQ_Checklist.pdf).

4. In the Methods/Analysis section the topic summaries are described as being “identified”. In Braun & Clarke’s 2019 paper on TA (https://www.tandfonline.com/doi/full/10.1080/2159676X.2019.1628806) they explain: “Themes are analytic outputs developed through and from the creative labour of our coding. They reflect considerable analytic ‘work,’ and are actively created by the researcher at the intersection of data, analytic process and subjectivity. Themes do not passively emerge from either data or coding; they are not ‘in’ the data, waiting to be identified and retrieved by the researcher.” Consider amending to terms like ‘developed’ ‘constructed’ or ‘generated’ to capture this process.

Overall this is an excellent manuscript and I recommend this for publication with minor revisions. I would like to extend my congratulations to the authors and patients involved.

6. PLOS authors have the option to publish the peer review history of their article (what does this mean?). If published, this will include your full peer review and any attached files.

Reviewer #1: No

Reviewer #2: **Yes: **Leanne Shearsmith

---

## [Author Response · Author response to Decision Letter 0]

9 Dec 2022

Responses to reviewers (SEE ALSO AS ATTACHED FILE)

We thank both reviewers for their comments and suggestions which have helped to improve the quality of this manuscript. 

We have provided a response below each comment using blue text. We provide reference to where additional text has been included in the revised manuscript and where this inclusion is placed. In addition, we have included two updated manuscripts, one with tracked changes as well as a ‘clean’ version. We have also added an additional supplementary document (S3 Appendix 3. COREQ checklist). We hope that you find our revised manuscript suitable for publication. 

Itemised response to reviewer comments

(Line numbers are accurate when viewing in simple mark-up)

The introduction provides a valuable account of patient activism during the pandemic, and provides an informative background and history of “Long COVID”’s identification. The points citing tweets from Elisa Perego (references 2 and 3, line 7-13) may be better supported by the publications she co-authored on the topic. See Callard, F., & Perego, E. (2021). How and why patients made Long Covid. Social science & medicine, 268, 113426. (https://www.sciencedirect.com/science/article/pii/S0277953620306456) and Munblit, D., O'Hara, M. E., Akrami, A., Perego, E., Olliaro, P., & Needham, D. M. (2022). Long COVID: aiming for a consensus. The Lancet Respiratory Medicine. (https://www.sciencedirect.com/science/article/pii/S2213260022001357)

We have added these references as suggested (lines 7-13 and references 2 and 3). The original choice of the tweets was made to try to reflect the online emergence of awareness around Long COVID but we agree with the review that the academic publications support our points better. 

 The authors’ reflections on diversity and inclusion strengthen this work. Given the impetus in the introduction to recruit a diverse sample and the reported lack of diversity in the discussion, it would be interesting to know the diversity of the REACT cohort the authors were able to select from. This could be expanded on in the discussion – was there a diverse sample in the REACT study but the pilot interview study failed to recruit a representative selection, or was the present study limited by an already non-diverse REACT cohort?

The REACT study had a representative sample however the pilot interview study was limited by its size and recruitment approach, so this was not reflected in the pilot sample. We have clarified this in lines 376-380.

Consider following reporting standards by including the COREQ (COnsolidated criteria for REporting Qualitative research) Checklist in your supplementary information. This is a checklist of items that should be included in reports of qualitative research (https://cdn.elsevier.com/promis_misc/ISSM_COREQ_Checklist.pdf).

We have used this checklist and made additions that were missing or not clear from the reporting. The checklist is presented in the supplementary material (S3 Appendix 3. COREQ checklist). The changes made are:

Line 58 we provided the specific months when data collection was carried out.

Line 68 we explicitly stated that we used a convenience sampling approach.

Line 72 and line 79 we specified which researchers carried out which aspects of data collection.

Line 77 we detailed whether we had numbers of those who declined to participate.

Line 98 we indicate the length of the interviews.

Line 103 we stated that researchers kept fieldnotes.

Line 107 we referenced the codebook or ‘code tree’ used by researcher.

Line 109 we specified whether preliminary findings were discussed with participants. 

Line 112 we explained that the findings are descriptive topic summaries

Line 114 we stated whether we sought to reach data saturation. 

Lines 124-137 we have added a reflexivity statement giving more information about the researchers who carried out interviews and analysis.

 In the Methods/Analysis section the topic summaries are described as being “identified”. In Braun & Clarke’s 2019 paper on TA (https://www.tandfonline.com/doi/full/10.1080/2159676X.2019.1628806) they explain: “Themes are analytic outputs developed through and from the creative labour of our coding. They reflect considerable analytic ‘work,’ and are actively created by the researcher at the intersection of data, analytic process and subjectivity. Themes do not passively emerge from either data or coding; they are not ‘in’ the data, waiting to be identified and retrieved by the researcher.” Consider amending to terms like ‘developed’ ‘constructed’ or ‘generated’ to capture this process.

We thank the reviewer for reminding us of this. We have changed ‘identified’ to ‘developed’ in line 108. 

Response to editor comments

1. We have checked the formatting of the manuscript against PLOS-ONE’s style guidance and ensured heading styles and fonts sizes are correct. 

2. We have updated the funding information for financial disclosure section as follows:

This work was supported by the NIHR/UKRI [grant number COV-LT-0040]. HW is a National Institute for Health Research (NIHR) Senior Investigator and acknowledges support from NIHR Biomedical Research Centre of Imperial College NHS Trust, NIHR School of Public Health Research, NIHR Applied Research Collaborative Northwest London and the Abdul Latif Jameel Institute for Disease and Emergency Analytics. GSC is supported by a National Institute for Health Research (NIHR) Professorship. PE is Director of the MRC Centre for Environment and Health (MR/L01341X/1, MR/S019669/1). PE acknowledges support from the NIHR Imperial Biomedical Research Centre and the NIHR HPRUs in Chemical and Radiation Threats and Hazards and in Environmental Exposures and Health, the British Heart Foundation Centre for Research Excellence at Imperial College London (RE/18/4/34215), Health Data Research UK (HDR UK) and the UK Dementia Research Institute at Imperial (MC_PC_17114). The principal investigator (PE) received the following awards: This study forms part of REACT Long COVID. 

We also acknowledge support from the NIHR Imperial Biomedical Research Centre, The Huo Family Foundation and the Department of Health and Social Care.

The funders had no role in study design, data collection and analysis ,decision to publish, or preparation of the manuscript. 

Competing interests: The authors have declared that no competing interests exist.

3. and 4. We have reviewed the PLOS-One guidance on data availability and find that there are ethical or legal restrictions on sharing a de-identified data set. The full study data cannot be made publicly available due to:

• The sensitive nature of this data (this is a small pilot study meaning there could be easy identification of participants), 

• The agreements and procedures governing usage and sharing of the collected dataset established by the study sponsors (UKRI/Medical Research Council and the National Institute for Health Research), research institution (Imperial College London) and Research Ethics Board (South-Central Berkshire B Research Ethics Committee (IRAS ID: 298404, REC Ref 21/SC/0134)).

• The privacy protection and the nature of the consent obtained from participants which ensured confidentiality and anonymity. 

However, de-identified participant data used in the present study may be made available to investigators who complete the following steps: 1) present a study proposal that has received approval from an independent research committee or research ethics board; 2) provide a data request for review by the study principal investigators (PIs) and co-investigators; 3) following approval of the request, execute a data-sharing agreement between the investigators and the Study PIs. Study proposals and data access requests for elements of raw data may be sent to the REACT Data Access Committee (contact via react.access@imperial.ac.uk)

Change to reference list:

On our final review of the manuscript, we noted a single incorrect reference which we have now corrected as follows:

Line 3-4 (Reference 1) was incorrectly attributed to:

Caroline Eccles. Creator of Message in a Bottle - Long Covid SOS film, Huckleberry Films 2020, Public involvement advisor feedback. 2021.

And now has been correctly attributed to:

Davis HE, Assaf GS, McCorkell L, Wei H, Low RJ, Re’em Y, et al. Characterizing long COVID in an international cohort: 7 months of symptoms and their impact. EClinicalMedicine [Internet]. 2021 Aug 1 [cited 2021 Oct 6];38. Available from: https://www.thelancet.com/journals/eclinm/article/PIIS2589-5370(21)00299-6/fulltext

The original reference referred to the support Caroline Eccles provided as a public advisor on this manuscript. This is now reflected by her being included as an author on this paper.

---

## [Decision Letter · Decision Letter 1]

12 Jan 2023

Awareness and perceptions of Long COVID among people in the REACT programme: early insights from a pilot interview study

PONE-D-22-18885R1

Dear Dr. Emily,

We’re pleased to inform you that your manuscript has been judged scientifically suitable for publication and will be formally accepted for publication once it meets all outstanding technical requirements.

Kind regards,

Nelsensius Klau Fauk, S.Fil., M., MHID, MSc, PhD

Academic Editor

PLOS ONE

Additional Editor Comments (optional):

Dear Authors, thanks for your detailed response to each comment from the reviewers. Your manuscript has improved significantly and is accepted for publication.

Reviewers' comments:

Reviewer's Responses to Questions

**Comments to the Author**

1. If the authors have adequately addressed your comments raised in a previous round of review and you feel that this manuscript is now acceptable for publication, you may indicate that here to bypass the “Comments to the Author” section, enter your conflict of interest statement in the “Confidential to Editor” section, and submit your "Accept" recommendation.

Reviewer #2: All comments have been addressed

2. Is the manuscript technically sound, and do the data support the conclusions?

Reviewer #2: (No Response)

3. Has the statistical analysis been performed appropriately and rigorously? 

Reviewer #2: (No Response)

4. Have the authors made all data underlying the findings in their manuscript fully available?

Reviewer #2: (No Response)

5. Is the manuscript presented in an intelligible fashion and written in standard English?

Reviewer #2: (No Response)

6. Review Comments to the Author

Reviewer #2: (No Response)

7. PLOS authors have the option to publish the peer review history of their article (what does this mean?). If published, this will include your full peer review and any attached files.

Reviewer #2: **Yes: **Leanne Shearsmith

---

## [Editor Report · Acceptance letter]

17 Jan 2023

PONE-D-22-18885R1 

Awareness and perceptions of Long COVID among people in the REACT programme: early insights from a pilot interview study 

Dear Dr. Cooper:

I'm pleased to inform you that your manuscript has been deemed suitable for publication in PLOS ONE. Congratulations! Your manuscript is now with our production department. 

Kind regards, 

on behalf of

Dr. Nelsensius Klau Fauk 

Academic Editor

PLOS ONE